# Human Milk Oligosaccharide Profiles and Associations with Maternal Nutritional Factors: A Scoping Review

**DOI:** 10.3390/nu13030965

**Published:** 2021-03-17

**Authors:** Caren Biddulph, Mark Holmes, Anna Kuballa, Peter S. W. Davies, Pieter Koorts, Roger J. Carter, Judith Maher

**Affiliations:** 1School of Health and Behavioural Sciences, University of the Sunshine Coast, Maroochydore DC, Queensland 4558, Australia; MHolmes@usc.edu.au (M.H.); akuballa@usc.edu.au (A.K.); jmaher@usc.edu.au (J.M.); 2Child Health Research Centre, University of Queensland (UQ), St Lucia, Queensland 4072, Australia; ps.davies@uq.edu.au; 3Department of Neonatology, Royal Brisbane and Women’s Hospital, Herston, Queensland 4029, Australia; pieter.koorts@health.qld.gov.au; 4Liaison Librarian, Science, Health, Nursing and Midwifery, and Sport Sciences, University of the Sunshine Coast, Maroochydore DC, Queensland 4558, Australia; rcarter@usc.edu.au

**Keywords:** maternal diet, maternal body composition, human milk oligosaccharide, breastfeeding

## Abstract

Human milk oligosaccharides (HMOs) are complex unconjugated glycans associated with positive infant health outcomes. This study has examined current knowledge of the effect of maternal diet and nutritional status on the composition of HMOs in breast milk. Using the PRISMA-ScR guidelines, a comprehensive, systematic literature search was conducted using Scopus, Web of Science, Global Health (CABI), and MEDLINE. Titles and abstracts were screened independently by two reviewers against predefined inclusion and exclusion criteria. Fourteen studies met the inclusion criteria and reported on maternal dietary intake (*n* = 3), maternal body composition indices (*n* = 9), and dietary supplementation interventions (*n* = 2). In total, data from 1388 lactating mothers (4011 milk samples) were included. Design methodologies varied substantially across studies, particularly for milk sample collection, HMO analysis, dietary and body composition assessment. Overall, this review has identified potential associations between maternal dietary intake and nutritional status and the HMO composition of human milk, though an abundance and sufficiency of evidence is lacking. Standardised procedures for human milk sample collection and HMO analysis, along with robust and validated nutrition assessment techniques, should be employed to further investigate the impact of maternal nutritional factors on HMO composition.

## 1. Introduction

Breast milk is a unique and ideal source of nutrition for most infants [1]. It is comprised of essential nutrients in the right proportions and contains beneficial bioactive factors such as antibodies, hormones and over 200 defined human milk oligosaccharides (HMOs) [1]. HMOs are structurally diverse and complex unconjugated glycans and represent the third largest solid component of human milk after lactose and lipids [2,3]. Their presence in human milk is associated with many positive infant health outcomes, and they play an important role in influencing the development of the infant immune system and gut microbiome [2,4]. In the infant intestinal tract, HMOs act as prebiotics and have selective anti-adhesive, antimicrobial properties [2,5]. As a potential dietary source of sialic acid, some of the sialylated HMOs may be important for infant learning and memory development, though this observation is based primarily on animal studies [6,7].

Whilst there has been significant research into the physiological and immunological properties of certain HMOs, little is currently understood about the maternal factors that affect the composition of oligosaccharides in human milk, particularly those not associated with genetic influences. HMO concentrations and composition in the milk of lactating women shows substantial intra- and inter-individual differences [8]. The total HMO content in milk decreases over the course of lactation, with colostrum containing 20–25 g/L and mature milk containing approximately 5–15 g/L [9]. Differential compositional changes also take place over time; levels of fucosylated HMOs are highest early in lactation and most other HMOs appear to decrease, or remain quite constant [8,10]. The variation in HMO composition is primarily due to maternal genetic factors, with numerous studies reporting on the effects of polymorphism of the Secretor and Lewis genes on the amount and diversity of HMOs [2,9,11]. Lactating women are often classified as Secretors or non-secretors based upon the presence or absence of the fucosyltransferase 2 (‘FUT2′) gene, with Secretor mothers producing more neutral α1-2-fucosylated HMOs in their milk [9]. Emerging research indicates that other interrelated and somewhat complex factors may influence the composition of HMOs [12], including parity and mode of delivery, environment influences such as geographic location and seasonality, and maternal influences such as age and nutritional status [12,13,14,15,16,17,18,19,20,21,22]. There is limited information and no synthesis on the effect of maternal nutritional intake, diet quality, and body composition on HMO composition during lactation [23].

Given the numerous and varied beneficial effects of HMOs on infant health and development [24,25], a better understanding of the influence of maternal dietary factors on HMO composition is desirable. The aim of this scoping review is to investigate any known associations between HMO composition and maternal nutrition and nutrition-related factors during lactation. The primary objective of this review is to assess the impact of maternal nutritional intake and patterns, nutritional status, and nutrition-related aspects, such as the maternal microbiome, on the composition of HMOs in breast milk. A brief discussion on hypotheses around possible mechanisms by which these factors may influence HMO profiles will be included. This review is intended for general clinicians, nutritionists, dietitians, neonatologists, metabolomics researchers, analytical chemists, and those working in lactation research and donor milk banks.

## 2. Materials and Methods

### 2.1. Study Design

This scoping review was planned and conducted using the Preferred Reporting Items for Systematic Reviews and Meta-Analyses guidelines extension for scoping reviews (PRIMSA-ScR) [26].

The review protocol was drafted using the “Preferred Reporting Items for Systematic Reviews and Meta-analysis Protocols” (PRISMA-P) and has been made publicly available; reference: Biddulph, C., Davies, P., Carter, R.J. and Maher, J. (2020). Human milk oligosaccharides profiles and associations with maternal nutritional factors: A scoping review protocol. USC Research Bank. https://research.usc.edu.au/discovery/delivery?vid=61USC_INST:ResearchRepository&repId=12126689620002621#13129825250002621 (accessed on 15 March 2021).

### 2.2. Identifying the Research Question

We aimed to address the following question: “What is known about the influence of maternal dietary intake, nutritional supplementation and body composition on the composition of HMOs in breastmilk?”.

### 2.3. Search Strategy and Eligibility Criteria

Two authors (C.B. and J.M.) performed concurrent and comprehensive systematic electronic literature searches to find relevant studies reporting on the links between maternal nutritional factors and HMOs. Recommended health and nutrition electronic databases were searched on 17 February 2020 at 13:00 AEST. These databases included: Scopus, Web of Science, Global Health (CABI) and MEDLINE (incl. PubMed). The following search strategy was applied to terms listed within the titles, abstracts, and keywords of articles:
(maternal OR mother OR human OR female) AND (diet* OR supplement* OR nutri* OR macronutrient* OR carbohydrate OR sugar* OR protein OR fat* OR fibre OR fiber OR prebiotic* OR probiotic* OR “body composition” OR bmi OR weight OR microbiome* OR bacteria* OR “entero-mammary pathway” OR lact*) AND (HMO OR “human milk oligosaccharide*” OR “breastmilk oligosaccharide”).

A research librarian (R.C.) assisted in advising on the search terms and eligibility criteria. The review includes original research papers that investigated associations between HMO content or profiles and at least one maternal factor that is nutrition-related, such as dietary intake, supplementation, nutritional status, and body composition. We selected studies in humans published as full-length articles and excluded conference abstracts, editorials, letters to the editor and case reports. We included randomised controlled trials, observational cohort studies, cross-sectional observational studies, and excluded reviews of mixed methods studies. Outcomes that were assessed included the total HMO content and absolute concentrations of single HMOs. There was no time exclusion on publication dates.

The final search results were exported into *Endnote^®^* reference management software, and duplicates were removed. The process was repeated for subsequent databases and sources, with articles sorted into folders and details captured as per the PRISMA flow chart (Figure 1). These results were supplemented by articles found using methods such as citation searching of relevant articles, snowballing, and reference list searching.

### 2.4. Article Screening and Data Abstraction

Two authors (C.B. and J.M.) screened the titles and abstracts of all papers and performed article selection independently according to the inclusion and exclusion criteria.

Data were extracted from the full-text papers by C.B. using a predefined data collection proforma, and subsequently reviewed by J.M. Information was collected on study characteristics and design, location, description of participants and sample size, milk sample collection, maternal nutrition assessment measures, HMO analysis, and main outcomes and conclusions. Data abstraction details are included in Table 1.

### 2.5. Quality of Reporting Assessment

In order to appraise the methodological quality of the studies, the Critical Appraisal Skills Programme (CASP) qualitative checklist tools were used, as endorsed by the Cochrane Qualitative and Implementation Methods Group (https://casp-uk.net/casp-tools-checklists/, accessed on 5 September 2020). Twelve studies were assessed using the Quality Assessment Tool for Observational Cohort and Cross-Sectional Studies, and two were assessed using the Quality Assessment of Controlled Intervention Studies tool [31]. Two reviewers (C.B. and J.M.) scored each manuscript independently and discussed results. A study was deemed to be of “good” (10–14 points), “fair” (5–9 points), or “poor” (<5 points) quality, based upon the validity of results and risk of bias. Critique ratings indicated that even in the case of “fair” scores, study results were valid despite being susceptible to some bias. We did not exclude studies based on quality as we aimed to canvas all results available for the scoping review.

## 3. Results

### 3.1. Synthesis

The search strategy identified a total of 2861 records (Figure 1). Three additional articles were identified from the reference lists of included publications. After removal of duplicates, 2137 remained. Based on title and abstract screening, 1698 articles were excluded. Of the remaining articles, 33 met the stated inclusion criteria and were subjected to full-text assessment. In total, 14 publications were included in the scoping review, with data from 1388 lactating women and 4011 milk samples. A summary evidence table of included studies is provided as Table 1. Papers reported on maternal dietary intake (*n* = 3), dietary supplementation interventions (*n* = 2), and maternal body composition (*n* = 9), in relation to HMO composition of milk samples. The articles reflect a quite recent interest in the link between maternal factors and HMO composition, being published between 2013 and 2020. Four studies were conducted North America [12,27,28,29], two in South America [15,30], three in Europe [8,19,22], two in Africa [14,18], one in Australia [16], and one in Asia [21]. One large multisite study with 11 international cohorts was also included [20].

### 3.2. Analysis of Methodologies

#### 3.2.1. Study Design and Sample Characteristics

Of the 14 studies included in the analysis, four were cross-sectional [20,21,29,30] and five were prospective observational [12,14,19,27,28] studies. Most studies investigated a range of variables, including maternal nutrition factors and their associations with HMO concentrations in the samples. Three studies were classified as prospective longitudinal studies with HMO-related outcomes assessed three to six times over the postpartum period. Changes in HMO profiles over time could therefore be reported upon [8,15,16]. Two studies were controlled experiments: a single (assessor)-blind, parallel group controlled multi-nutrient supplementation trial [18], and a double-blind, placebo-controlled study with two parallel groups investigating maternal probiotic supplementation [22]. Seven of the studies (50%) were deemed to be of good quality [12,18,19,20,22,28,30] and half received a “fair” rating [8,14,15,16,21,29,30]. Both controlled intervention studies were rated highly, with adequate randomisation and intention-to-treat analyses [18,22]. The sample size varied substantially between studies, ranging from pilot/proof of concept (*n* = 20 mother-infant pairs) [16] to a large multi-national trial (*n* = 410 across 11 sites) [20]. The largest study with a sample size of *n* = 427 women [12] was embedded within a larger observational study, the Canadian Healthy Infant Longitudinal Development (CHILD) cohort which recruited 3624 pregnant women [32]. Typically, the sample size was pragmatically chosen and many of the studies investigating dietary parameters and HMOs did not report on the strength of correlational relationships. Since work in this field is still very much exploratory in nature, the level of variance of interest from a clinical perspective remains uncertain. Most of the studies considered basic exposure variables such as: maternal age, ethnicity, socioeconomic status, geographical location, health status, gestational age, parity, birth weight, mode of delivery, and breastfeeding status. Few studies (*n* = 3) investigated the use of nutritional supplements (including herbal and pre-/probiotics), and only one study used body composition indices other than the standard calculation of body mass index (BMI) [16].

#### 3.2.2. Human Milk Sampling Procedures

Milk sampling protocols varied between studies, reflecting the range of methodologies applied in the field. Most studies (*n* = 11) measured the HMO composition in mature milk samples However, one study measured HMOs in stored frozen colostrum samples (0 to 4 days postpartum) [22] and two studies included samples of transitional milk at 28 days [8] and 3 days postpartum [15]. To assess trajectories and account for the fact that HMO profiles vary throughout the course of lactation [9], some studies sampled milk at multiple time points [14,15,16,19,20,28], whilst others only included a single sample of mature milk in their design [12,18,21,27,29]. Of the papers that reported the time of day of sampling, most indicated this was done either in the morning [14,15,27,30] or around midday [8,28]. Only four studies reported on HMO composition in combined fore- and hind-milk from multiple feeds over 24 h [12,29] or pooled samples of both breasts [14,19]. Substantial sources of variance are evident across the studies with regards to breast milk sampling and handling protocols, time of day of expression, partial vs full breast expression, and the subsequent storage of samples prior to analysis. While evidence suggests that the HMO composition in mature milk is not affected by cold storage and seems to remain stable over and between days [33], the effect of long-term storage and multiple handling procedures on HMO composition remains unknown [1].

#### 3.2.3. Human Milk Oligosaccharides Analysis

For the analysis and profiling of HMOs, a variety of chromatographic, electrophoretic, and spectrometric approaches were used. Most of the studies used quantification methods based on liquid chromatography (LC) coupled to mass spectrometric (MS) detection [12,14,18,19,20,22,29,30]. Two studies used capillary electrophoresis for HMO profiling in addition to this [8,27]. A further two utilised high performance LC (HPLC) with fluorescence detection [15,21], and one used LC-GCMS (liquid and gas chromatography) for untargeted metabolomics analysis [28]. The impact of varying HMO analysis methods on study outcomes is difficult to ascertain; this would require the analysis of the same samples across different laboratories and methods for assessment of variance. There is currently no gold standard method for HMO analysis, and variances across studies and laboratories remain.

Most of the studies sought to quantify at least 16–20 individual HMOs that typically account for > 90% of total HMO content [12], and their concentrations were summed to estimate total HMO concentrations [8,12,14,15,19,20,22,29,30]. The following HMOs were often summed to provide an estimate of total HMO content in milk samples: 2′-fucosyllactose (2′-FL), 3-fucosyllactose (3FL), 3′-Sialyllactose (3′SL), 6′-Sialyllactose (6′SL), lacto-N-tetraose (LNT), lacto-N-neotetraose (LNnT), lacto-N-fucopentaose I/II/III (LNFP I, LNFP II, LNFP III), sialyl-LNT (LST) b, LSTc, difucosyl-LNT (DFLNT), disialyl-LNT (DSLNT), fucosyl-lacto-N-hexaose (FLNH), difucosyl-lacto-N-hexaose (DFLNH), fucosyl-disialyl-lacto-N-hexaose (FDSLNH), disialyl-lacto-N-hexaose (DSLNH), difucosyllactose (DFLac), and lacto-N-hexaose (LNH). Two studies did not report on HMO quantification as such, and instead reported on a calculated total HMO value estimated form the total milk carbohydrate and lactose content [16], and on milk sialic acid concentrations as a proxy for sialylated HMO content [21]. HMO concentrations are strongly influenced by genetics [12], thus most of the studies assessed maternal Secretor status as a confounding variable [8,12,14,15,19,20,22,27,29,30]. Secretor can be determined by genotyping the presence of *FUT2* genes via polymerase chain reaction-random fragment length polymorphisms [34]. However, studies in this review used a proxy method involving HMO identification and the presence or abundance of fucosylated HMOs such as 2′-FL and LNFP I [12,15,19,20,29,30]. Davis et al. chose a relative cutoff of > 6% relative α (1-2) fucosylation to be indicative of Secretor status [14]. Only one paper reported both Secretor and Lewis phenotypes based on the abundance or lack of α1-2 and α1-4 fucosylated HMOs [31].

#### 3.2.4. Nutrition Assessment

*Dietary intake assessment methods.* Maternal dietary intake was assessed directly in three studies [12,21,27]. Validated food frequency questionnaires (FFQ) were used in two studies [12,21], where the usual frequency of consumption of foods and beverage were assessed retrospectively. Azad et al. [12] used the *Women’s Health Initiative (USA)* FFQ during late pregnancy (not during lactation) and assessed pre-pregnancy BMI using self-reported weight. This FFQ is validated in non-pregnant women, captures usual dietary intake, and was used to calculate the Healthy Eating Index (HEI) score, resulting in an overall score that represents the quality of the whole diet as compared to recommendations of the Dietary Guidelines for Americans [12]. A more recent study by Quin et al. [27] utilised the Self-Administered 24 h (ASA24) Dietary Assessment tool [35], which was completed by 16 lactating women in the 24-h prior to milk sample collection. This meant that the nutrition assessment was based on the mother’s dietary intake immediately prior to milk expression during lactation. In contrast, the FFQ tool utilised by Azad et al. [12] assessed longer term or usual maternal dietary intake during pregnancy. Both studies [12,30] represented dietary intake as food groups and reported on macronutrient profiles (total energy, fat, protein, and carbohydrates). The third study by Qiao et al. [21] also used an FFQ that included selected micronutrients (vitamins A, E, C, calcium, zinc, manganese, iron, and selenium). This study sought to investigate the impact of maternal diet on breastmilk components, specifically sialic acid. HMOs were not quantified; rather breastmilk sialic acid levels (of which 82% was bound to oligosaccharides) were measured, and the 72-h FFQ was performed during lactation (40 ± 7 days postpartum). This FFQ assessed participants’ recent dietary intake relative to milk sampling (single occurrence) and food models were used to improve the accuracy of estimation of dietary intake [21]. Other dietary exposures or patterns were not explored in these three studies. Dietary intake was not directly measured in a probiotic supplementation trial [22] and a multi-nutrient supplement trial [18]. In these intervention studies, adherence to protocol indicating actual dietary intake of supplements was assessed using developed questionnaires.

*Anthropometric assessment methods.* Eleven of the fourteen studies assessed maternal anthropometric characteristics, including BMI, gestational weight gain, and body composition. These anthropometric measures can be used as a proxy or indicator of both short- and longer-term nutritional status. Basic maternal anthropometric measures of height, body weight/mass and BMI were assessed in 11 of the studies [8,12,14,15,16,18,19,20,28,29,30]. BMI was calculated and classified using the standard equation: Weight/Height^2^ (Quetelet’s index; kg/m^2^) [36]. However, BMI is a crude measure and cannot be used to distinguish between fat and lean mass [37]. Five studies assessed both pre-pregnancy BMI and BMI during lactation [8,18,19,28,30], though key confounders of BMI such as levels of physical activity, dietary intake and hormonal influences were not accounted for. Gestational weight gain (GWG) was determined in four studies as the difference between body weight at the final prenatal visit [15,19,28] and pre-pregnancy weight (self-reported or from medical records). Only Gridneva et al. [38] used bioelectrical impedance spectroscopy (BIS) on four occasions (2, 5, 9, and/or 12 months postpartum) in addition to body weight and BMI to assess maternal body composition during lactation. BIS is considered a more robust technique of assessing body composition than BMI.

### 3.3. Maternal Diet Quality and Dietary Components

When overall pregnancy diet quality was assessed by Azad et al. [12] in their large general population–based cohort study (*n* = 427) no associations with HMO concentrations in milk collected at 3–4 months post-partum were found. However, total energy intake, although not associated with overall HMO concentrations, was positively correlated with some individual HMOs (LNTand DFLNT) (all *ρ* = −0.11 to +0.1) (12). Davis et al. [14] found that in the so-called ‘dry’ season in African Gambia, when food is more plentiful, mothers could produce significantly more HMOs. Season was described as a proxy for dietary energy intake. It was thought that as total dietary energy intake may be greater in the dry season, the overall milk output may be increased. However, no direct assessments were made of the mothers’ dietary intake so the link between HMO composition and seasonal environmental changes as a proxy for dietary energy intake in this study remains speculative. Azad et al. [12] also reported on seasonal variation in some HMOs but suggested that environmental factors such as climate or exposure to sunlight or allergens might influence HMO concentrations more than dietary energy intake.

Variable associations between maternal dietary macronutrient intakes and HMO concentrations were found in three studies [12,21,27]. Azad et al. [12] found weak negative correlations between total protein, empty calories, and sialyl-lacto-N-tetraose b (LSTb). Quin et al. [27] also reported that some maternal dietary components could influence the biosynthesis of HMOs during lactation, particularly in Secretor-positive women. Several sulfonated HMOs were positively correlated with monounsaturated and polyunsaturated fats, and negatively correlated with levels of saturated fats and dietary cholesterol (*p* < 0.05). Both studies also explored the potential association between food groups and HMO composition in the milk samples. Quin et al. [27] found that only fruit, a dietary source of simple sugars and dietary fiber, was positively correlated with absolute amounts of selected HMOs and with levels of galactose and fucose present in HMOs, while sialic acid levels were significantly lower (*p* < 0.05) [30]. On the other hand, Azad et al. [12] found a positive correlation between wholegrains and fucosyllacto-*N*-hexaose. No other significant relationships between food groups or dietary patterns and HMO concentrations were reported.

In terms of micronutrient content of the maternal diet, evidence is also limited at this stage. In the study by Qiao et al. [21], mothers who had higher dietary vitamin A intakes (from food sources; 602.22 ± 126.46 µg/day) had a significantly higher concentration of sialic acid in their milk samples (*p* = 0.000). No significant associations for any other dietary components were noted, although profiling individual HMOs at one time point only is a limitation of the cross-sectional method used. Repeating dietary assessment over time may give more insight into the longitudinal changes in HMO profiles over the course of lactation. Three of the studies investigated associations between dietary supplementation (nutrient or probiotic) and HMO composition [12,18,22]. Azad et al. found that only DSLNH was notably higher in the breast milk of mothers who self-reported taking multivitamins supplements [12]. In a large-scale interventional trial, Jorgensen et al. [18] hypothesised that nutrient supplementation during late pregnancy would impact HMO levels, presumably secondary to improving overall nutritional status. However, when assessing a cohort of mothers in Africa, most of whom had suboptimal nutritional intake, they found no effect of a lipid-based nutrient supplements on HMO levels (*p* > 0.10 for all comparisons) [18]. Pregnant women (*n* = 647) were given either micronutrient capsules containing 18 micronutrients or a 20g-dose of a high-energy, micronutrient fortified lipid-based nutrient supplement with the same 18 micronutrients and 4 additional minerals, 2.6 g protein, and 10 g fat to provide 118 kcal of total energy. Neither intervention resulted in increased HMO levels at 6 months postpartum. However, no investigation was made into the adequacy or quality of the mothers’ diet, so other dietary influences could not be discounted. Since participants in this study were often of a lower socioeconomic status and with low BMIs, it is possible that the effects of additional nutrient supplementation may only be seen in energy-replete, well-nourished individuals.

One study reported on the impact of probiotic supplementation on HMO profiles [22], however their results reported on a complex interplay between the maternal microbiome, the milk microbiome, and HMOs, which made it difficult to report directly on the relationship between probiotic supplementation and HMOs. In this randomized, double-blind, placebo-controlled study probiotic supplements were given to mothers (*n* = 81) during the late stages of pregnancy. The mothers in the probiotic group took twice daily capsules containing *Lactobacillus rhamnosus GG*, *Lactobacillus rhamnosus LC705*, *Bifidobacterium breve Bb99*, *and Propionibacterium freudenreichii ssp. shermanii JS*. Their newborn infants received an opened capsule containing the same probiotics mixed with 20 drops of sugar syrup containing 0.8 g of galacto-oligsaccharides once daily for 6 months after birth. Concentrations of 3-fucosyllactose (3FL) and 3′-sialyllactose (3′SL) in colostrum from the mothers in the probiotic supplementation group increased significantly compared with those in the placebo group (*p* = 0.008 and *p* = 0.006, respectively). An opposing and significant decrease in 6′-sialylated HMOs was also noted (*p* < 0.05 for all). It is not clear whether the changes in 3FL and 3′SL concentrations were specific to the types of probiotics used in this study, but it does suggest that external factors can manipulate HMO profiles. The investigators further suggested that probiotic supplements could influence the amount or activity of enzymes that are functional in specific biological pathways [22].

Overall, studies assessing maternal dietary intake or supplementation and the association with HMO volume and profiles are scarce. The variance in the methods of assessment is too great to allow for general conclusions. Dietary intake and quality in pregnancy does not appear to be of notable influence [12], whereas some individual dietary components and supplements may be influential in the short term or during the postpartum period [12,21,22,27]. The majority of the included studies had small samples which also limits the ability to draw any significant conclusions. Assessment of lactational diet through multiple dietary recalls may assist in improving validity of dietary methods. In addition, other potentially more robust methods for exploring diet quality and dietary patterns through principal component analyses and other similar methods, have not been explored.

### 3.4. Body Composition Assessment

A number of measures performed at various times in relation to lactation and considering different genetic subgroups exist in the literature examining maternal anthropometry and HMO composition. When assessing pre-pregnancy BMI and maternal body weight, Ferreira et al. [15] found that both measures were moderately correlated with LNnT (Spearman rank correlation 0.4) but Samuel et al. [8] found that a higher BMI was associated with a lower concentration of this same HMO and some others (LNT and LNFP V; all *p* < 0.05). The studies were however performed on different cohorts (South American versus European) and the substantial influence of genetic variation or geographic location may have outweighed the influence of maternal physiological status. The association noted in the latter study was also inconsequential after statistical correction for multiple testing, and then only 3′SL and 6′GL remained positively correlated with a higher BMI category over the first four months of lactation (*p* < 0.05) [8]. Both Secretor and non-secretor mothers were included in the analysis in these studies. Larsson et al. [19] found no associations between HMOs and pre- or post-pregnancy BMI, nor with gestational weight gain in Secretor positive women. However, when combining Secretors and non-secretors, maternal pre-pregnancy BMI was weakly associated with some individual HMO values. These same associations became significant with BMI measured at 5 months postpartum (i.e., negative associations with 6′-SL and LSTb (*p* = 0.03), positively with 2′-FL (*p* < 0.05 for all). Total HMO volume and total HMO-bound fucose were also positively associated with postpartum BMI (*p* = 0.015 and *p* = 0.033, respectively). The timing of anthropometric measurements relative to pregnancy and lactation appears to be of consequence. Another example of this impression is LNFP III: one study reported a moderately negative correlation for this HMO with pre-pregnancy body weight (r = −0.04) [15], another reported a positive correlation (r = 0.20) with postpartum body weight [20].

The association between anthropometric measures and individual HMOs appears to be more consistent when only considering postpartum assessments. McGuire et al. [20] performed all measurements during the lactational period and found correlations between maternal weight, BMI, and individual HMOs (positive correlations with 2′-FL (r = 0.20 for both) and FLNH (r = 0.19 and 0.15, respectively), and inverse correlations with DSLNT (r = 20.20 and 20.24) and LNnT (r = 20.16 and 20.21). Likewise, Tonon et al. [31] measured maternal BMI in the early postpartum period (17 to 76 days, median: 32 days, IQR: 25–46 days), and found a positive correlation with 2′-FL (r = 0.30, *p* < 0.05). As this was a cross-sectional study, the persistence of the association over time could not be determined. At one month postpartum, Isganaitis et al. [28] found that maternal BMI was significantly associated with 2′-FL, LNFP I, and LNFP II/III (*p* < 0.05 for all), but these associations did not extend to six months postpartum.

Overall, more evidence is required to confirm the associations between maternal anthropometrics and singular HMOs. Perhaps the most evident association across studies and populations is indeed for 2′-FL positively correlating with BMI in the postpartum period [19,20,28,30]. Future studies would do well to separate results for Secretor and non-secretor women. Further, body weight and BMI are not good indicators of adiposity during the postpartum period [20] and other more valid techniques should be used to investigate the associations more definitively. Gridneva et al. [16] used BIS on four occasions (2, 5, 9, and/or 12 months postpartum) in addition to maternal body weight and BMI measures. They found no associations with maternal body composition (fat mass, fat-free mass, percentage fat mass, and height-normalized indices of fat mass and fat-free mass) and calculated HMO levels (*n* = 20). The authors did not quantitate individual HMOs and perhaps direct identification of HMOs might have better revealed any associations between indices of body composition and HMO profiles during lactation.

## 4. Discussion

The aim of this scoping review was to investigate any known associations between maternal nutritional intake and patterns, as well as nutritional status and the volume and/or composition of HMOs in human milk. We also sought to investigate hypotheses around possible mechanisms by which maternal diet and related factors may influence HMO profiles. Work in this field is still in the initial exploratory stages, with most publications emerging in the last three years. Following the PRISMA-ScR guidelines and a systematic search of the literature, fourteen studies met the inclusion criteria. Only three reported on maternal dietary intake, nine on maternal body composition, and two on controlled dietary supplementation interventions. In total, data from 1388 lactating mothers and 4011 milk samples were included. The review identified singular associations between the HMO composition of human milk and components of the maternal diet or body composition measures.

Variation was introduced via interlaboratory differences in the protocols and procedures utilised for milk sampling and HMO analyses, which could impact the HMO quantities and profiles reported. However, most studies included in this review used chromatographic and mass spectrometric methods consistent with current practices to identify and quantify a select group of more abundant HMOs. The existence of published data and analytical standards was likely to also impact on the choice of HMOs examined [9,30]. Several studies focused on quantifying less than 20% of more than 200 structurally distinct and known HMOs [39] and accounted for the major source of interindividual variation in HMO levels; that is the effects of polymorphism of the Secretor and Lewis genes [2,11,30]. Results should be reported separately for Secretor and non-secretor women, although there is notable variability in HMO profiles even within the same phenotypes [30]. Discrepancies exist in the literature as to whether differences in maternal secretor status affect infant outcomes: evidence suggests no effect on growth outcomes [30], yet some effects on infant microbiota [40]. Future studies should also calculate the absolute amount of milk (and therefore HMOs) ingested by the infant, as certain health outcomes, such as growth rate, appear to be related to the volume of breastmilk consumed [16].

Research into the influence of maternal dietary intake on HMO composition and profiles in colostrum and mature milk is in its infancy and has been explored in only three studies included in this review [12,21,27]. Significant variability exists in dietary intake assessment methods, as reported above. Diet as an exposure is difficult to measure, due in part to the complexity and number of dietary elements that must be accounted for. Although diet quality has been assessed using a robust measure (HEI) [12], other methods for assessing diet quality such as dietary pattern analyses, could be used to further explore relationships between diet and HMOs. Further, diet quality was only assessed during pregnancy and should be assessed during lactation, preferably at multiple timepoints, as it is likely that maternal diet changes post pregnancy. The literature points to a decrease in overall diet quality (along with fruit and vegetable intake), and an increase in the intake of energy-dense yet nutrient-poor food choices post pregnancy [41].

The fact that HMO profiles also varies across lactation gives further reason to perform multiple dietary assessments in the postpartum period. Multiple 24-h dietary recalls may be more valid and reliable and provide an acceptable and more practical alternative to a weighed food diary. Quin et al. [27] used a single dietary recall in the postpartum period, in a sample too small to assess diet quality. As such their work was centered on dietary components alone but did reveal correlations between diet (total sugars, dietary fibre) and the fucose/galactose in HMOs. They suggest maternal diet is important in the biosynthesis of HMOs, which is likely since the biosynthetic pathway initiates from activated monosaccharides. This suggestion may be echoed in very recent work by Seferovic et al. [42] who found that changing maternal dietary energy and in particular, carbohydrate sources, resulted in alterations in some of the major HMO constituents in a short duration of time. In a cross-over controlled feeding study (*n* = 7 lactating women), a high fat diet (contributing > 40% total energy) resulted in a decrease in the concentration of sialylated HMOs, and a higher glucose versus galactose diet affected the profiles of fucosylated HMOs. These studies are expensive to implement and have limited generalisability, partly due to the requirement that participants receive controlled diets but do reveal mechanistic pathways in a clinical sense. Larger, longer duration intervention studies may reveal additional dietary components that impact the HMO composition of milk, as well as provide further evidence in identifying the underlying mechanisms driving such changes. However, these types of studies do need to be complemented with population-based studies to generate practical dietary advice to lactating women. Presently, the relationship between dietary manipulations and HMOs remains speculative due to the lack of a clear understanding of the pathways of HMO biosynthesis and because human milk supplies infant needs in a complex and dynamic manner. This review found no consistent relationships between HMOs and maternal dietary exposures in general populations. In fact, it is generally accepted that gross milk composition may be safeguarded against moderate variations in maternal nutritional status [43] and that suboptimal dietary (in particular, energy) intake of mothers does not have notable effects human milk macronutrient content [44]. Whether this is the case for maternal dietary macronutrient content or dietary patterns and HMO composition remains unknown.

Studies of dietary supplementation or specific micronutrients and their associations with HMO composition and profiles are also scarce, with two interventional studies included in this review [18,22]. At the very least, future trials should aim to include baseline measures and control for other sources of variance such as maternal and infant diets. When investigating pre/probiotic or symbiotic supplements, assessment of the interplay between maternal and infant gut microbiomes, as well as the milk microbiome might be valuable and relevant. Seppo et al. [22] found that probiotic supplements given during late stages of pregnancy changed the relative HMO composition, possibly by shaping the maternal gut microbiome. It is plausible that this in turn shapes the milk microbiome (via the entero-mammary pathway) and successively the overall milk environment of which HMOs form a part [13]. A possible diet-related mechanism may also be that diets high in fibre-rich food groups such as fruit and wholegrains may also shape the maternal microbiome. The work by the Azad [12] and Quin [27] groups found evidence of associations between both food groups and HMO composition, and perhaps the mechanism is the interplay between the maternal diet, gut microbiome, the milk microbiome, and the HMO composition therein. Certainly, initial evidence exists for a link between maternal dietary changes, HMOs, and changes in the milk microbiome metagenomic functional capacity [42].

The evidence for associations between maternal body composition and HMO profiles may be slightly more compelling. The hypotheses behind the reported associations vary and are speculative at this stage, but center around the idea that glycosylation (and therefore HMO composition) may be influenced by maternal physiological status. The largest multi-site, international study included in this review [20] confirmed correlations between maternal body weight and/or BMI and select HMOs during lactation. Even pre-pregnancy BMI and weight showed associations with HMOs [8,15], and these may be correlated with breastfeeding practices and the overall duration of breastfeeding [45]. Direct evidence linking maternal body composition and HMO profiles from the included studies remains limited however, and in many cases involved only singular changes in a few HMOs which differed across studies. It should also be noted that BMI does not seem to relate to overall breast milk quality and quantity, except in cases of severe undernutrition [46]. Further, the validity of self-reported pre-pregnancy body weight and the appropriateness of BMI as a measure of body composition during lactation is questionable. The need for standardised body composition tools beyond the BMI is clear. These tools would also need to be accurate, valid, reliable, and chosen after consideration of factors such as suitability/acceptability and cost. Measures such as dual-energy X-ray absorptiometry (DXA) and bioimpedance analysis (BIA) represent such options [47]. Despite being the current gold standard for body composition measurement, DXA was not used by any of the studies included in this review. The only study that utilised bioelectrical impedance spectroscopy (BIS) found no associations between maternal body composition and calculated HMO levels [16]. HMOs were not quantitated directly however, limiting investigations into the effect on singular or classes of HMOs. The discrepancies between, and paucity of results, certainly warrants further investigation into how maternal pre-pregnancy BMI, pregnancy weight gain, and maternal nutritional status during lactation affect HMO composition. The link between maternal dietary intake, including diet quality and dietary patterns, and body composition as they relate to HMO profiles, has also not yet been explored.

Overall, the findings from the studies included in this scoping review cautiously suggest that maternal dietary intake and body composition could affect the biosynthesis of HMOs, although this hypothesis and the mechanistic links would need to be confirmed in additional studies. One such concept to be explored is the interaction between maternal nutrition, its impact on HMO composition and the milk microbiome, and subsequently on the structure and function of the infant gut microbiome. Quin and colleagues [27] demonstrated that total dietary intake of sugars and saturated fats had a significant impact on HMO constituents and showed an additional correlation between HMO profile and infant gut microbial composition. Furthermore, a study linking maternal secretor status, HMO composition and the faecal microbiota of breastfed children show that although bacterial diversity is not affected by secretor status, the relative abundance of certain bacteria is affected [40]. A more detailed assessment of nutrient intake during lactation in particular may be required to identify (or exclude) dietary effects on HMO composition, preferably using quantitative methods of HMO analysis, valid and reliable nutrition assessment tools, and large samples of healthy mothers across different geographic settings. Little is known about the link between maternal physiological status (nutritional status in particular), and the interindividual variability of HMO levels. Further, the clinical significance of changes in HMO composition in terms of the impact on infant health (or disease) and development requires further investigation. It is hoped that modulating the HMO composition of human milk via modifications of the maternal diet, body composition and even the microbiome, may lead to opportunities to maximise health outcomes of breastfed infants. Potential future projects could include integrative exploration of the link between the maternal, milk, and infant microbiomes, extended therapeutic uses of HMOs, and the estimation of optimal levels of HMOs in human milk and formulas for infant health.

### Strengths and Limitations

This scoping review is the first to map what is known about maternal nutritional intake and status and the profile of HMOs, and it briefly assesses the methodological and analytical approaches in the included studies. Selection bias may have been introduced because only original journal articles published in English were included. Further, the review does not include articles published after March 2020 and examines only a limited number of available scientific papers. Despite these limitations, this review is strengthened by its compliance to the PRISMA-ScR guidelines and robust search strategy conducted in multiple databases.

## 5. Conclusions

The purpose of this review was to describe the extent of research undertaken to assess the effect of maternal dietary intake and body composition on HMO variability and content during established lactation. As these factors are largely modifiable, it is anticipated that the outcomes of this review will help to inform nutritional guidelines for lactating women in order to optimise the composition of bioactive oligosaccharides in human milk. Future studies should aim to follow standardised methods of milk sampling, storage, and analysis, as well as utilise robust nutritional assessment tools. Given the emerging and significant role that human milk plays in overall infant health and development, research into ways to optimise its nutritional superiority and HMO quality must continue to be a focus area.

### Update

At the time of this review, no study had clearly demonstrated associations between the maternal diet, HMO composition, and the breast milk microbiome. However, a recent paper has provided some evidence in this regard, albeit with small study numbers. In a series of inpatient, single-blinded cross-over dietary intervention studies, small groups (*n* = 7) of lactating women received diets that varied according to primary carbohydrate or dietary macronutrient source. Within even a short duration of intervention, results suggest that maternal diet can modulate the profile of HMOs as well as the functional capacity, though not taxonomic composition of the milk microbiome [42]. Further evidence is sought through larger long-term dietary studies.

## Figures and Tables

**Figure 1 nutrients-13-00965-f001:**
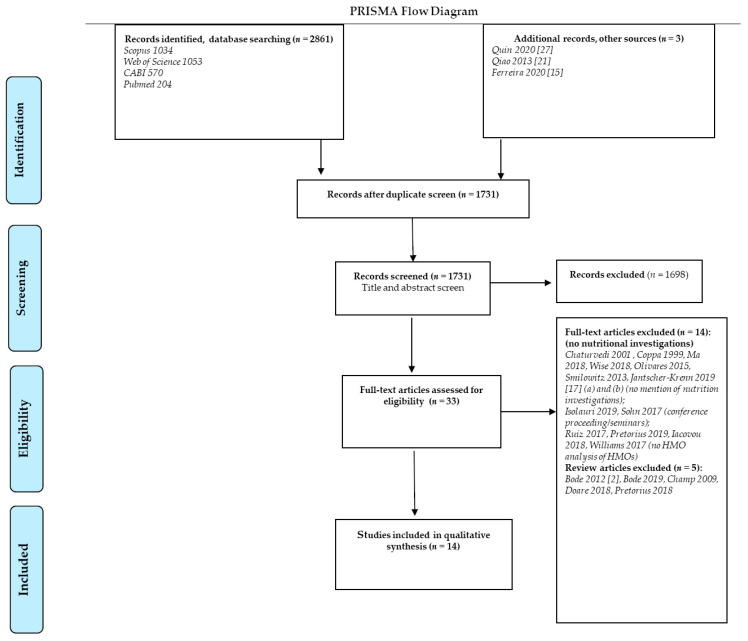
Flow diagram describing study selection process [2,15,17,21,27].

**Table 1 nutrients-13-00965-t001:** Studies evaluating the possible effects of maternal nutrition-related factors on HMO content and profiles.

Reference	Exposure Variables of Interest	Study Design	Population and Sample Number	Maternal Nutritional Assessment Tool	Milk Sampling	HMO Analysis	Relevant Main Outcomes
Azad et al. [12]	Maternal pre-pregnancy BMI and usual dietary intake during pregnancy.	Prospective observational study.	*n* = 427 milk samples, 1 from each mother-infant dyad (healthy term infants, Canadian multi-ethnic mothers).	Validated FFQ during pregnancy; pre-pregnancy BMI (self-reported weight, measured height).	Combined and refrigerated single sample from multiple feeds during a 24-h period at 3–4 months postpartum (median: 16 weeks, IQR: 14–19 weeks).	HPLC; 19 HMOs; total and relative abundance of HMOs; total HMO-bound fucose and sialic acid; FUT2 “Secretor status” defined by 2′-FL.	Maternal dietary intake and BMI not correlated with HMO concentrations.
Davis et al. [14]	Seasonal environmental changes as proxy for caloric intake.	Prospective observational study.	*n* = 99 milk samples, from 33 mother-infant dyads (rural African Gambian women).	Maternal body weight. Season/environment as proxy for caloric intake.	33 samples × 3 time points (postpartum weeks 4, 16, 20); 5mL of hand-expressed milk from each breast into a separate tube in the morning. Start, middle or end of feed not specified.	Nano-HPLC-Chip/TOF Mass Spectrometry; 19 HMOs. Total HMOs, Fucosylated, sialylated, undecorated HMOs. Mothers with less than Secretors > 6% relative α (1–2) fucosylation.	HMO levels are associated with season. HMO concentrations/total amount significantly higher (*p* = 0.01) in dry season (food is more plentiful and hence total dietary energy intake may be greater).
Ferreira et al. [15]	Maternal anthropometric characteristics.	Prospective longitudinal observational study.	*n* = 174 milk samples from 101 subjects (healthy, Latin-American, pregnant women).	Pre-pregnancy BMI (self-reported pre-pregnancy weight and measured height), gestational weight gain (weighed at last prenatal visit); Supplement use via questionnaire (iron and folic acid).	Manual expression of milk samples at 2–8 days, 28–50 and 88–119 days, in the morning after breakfast.	HPLC with fluorescence detection (HPLC-FL); 19 HMOs. HMO absolute (total) and relative abundances. Secretor status determined by presence of 2′-FL and LNFP I.	Maternal pre-pregnancy weight and BMI is associated with HMO composition (low-moderate Spearman correlation values; positively correlated with LNnT (0.4) and inversely correlated with LNFP III (-0.4)).
Gridneva et al. [16]	Maternal body composition indices (fat-free and fat mass, percentage fat mass (%FM).	Longitudinal proof of concept/pilot study.	*n* = 80 milk samples, from 20 mother-infant dyads (healthy term infants and Australian, predominantly Caucasian mothers).	Maternal body weight, BMI, and body composition indices at 4 visits utilising BIS.	Small (1–2 mL) pre-/post feed milk samples were collected into 5 mL polypropylene vials when infants were 2 and/or 5, 9, and 12 months-old.	None (Total HMO concentration (g/L) estimated by deducting lactose concentration from total carbohydrate concentration). No determination of Secretor status.	No associations with maternal body composition; prior to FDR adjustment, higher HMO calculated daily intake was associated with higher maternal %FM and FM/FFM between 2 and 5 months, and with lower maternal %FM and FM/FFM at 9 and 12 months.
Isganaitiset al. [28]	Maternal obesity and postnatal weight gain.	Prospective observational study.	*n* = 57 total milk samples; from 31 mothers at 1 month and 26 at 6 months (American, predominantly Caucasian women).	Maternal weight, height, BMI at study site by researchers. Participants grouped by maternal pre-pregnancy BMI (*n* = 15 BMI *<* 25, lean, *n* = 16 BMI ≥ 25 kg/m^2^, overweight/obese). Gestational weight gain, BMI at 1 month postpartum.	Complete expression of a single breast (right preferred) at 1 and 6 months postpartum using an electric pump. 2–2.5 h after last feed between 8 and 10 am (around midday).	Untargeted metabolomics analysis using LC-GCMS. Reported on 3 oligosaccharides: 2′-FL, LNFP I, and LNFP II/III. No determination of Secretor status.	Maternal obesity was linked to differences in HMO composition at 1 month postpartum (2′-FL, LNFP I and LNFP II/III significantly correlated with maternal BMI).
Jorgensenet al. [18]	Lipid-based Nutrient (LNS) or multiple micronutrient (MMN) supplements, compared with iron and folic acid during pregnancy and placebo postpartum. Potential covariates included baseline maternal BMI (in kg/m2).	Randomised, single (assessor)-blind, parallel group-controlled supplementation trial. Outcomes assessed according to intention to treat principle.	*n* = 645 breastmilk samples (rural Malawian (African) women, low socio-economic status).	Adherence index for supplement compliance. Weight and height in triplicate by trained anthropometrists at enrollment to calculate BMI during pregnancy (≤20, at 32 and 36 gestational weeks) and once after birth (at 1–2 weeks after delivery).	Single sample, manual expression of the full content of one breast into a sterile plastic cup at 6 months postpartum.	Nano-LC microfluidic chip coupled to electrospray time-of flight mass spectrometer. Summed total HMOs, fucosylated, sialylated and nonfucosylated neutral glycans. No determination of Secretor status.	Supplementation with an LNS or MMN capsule during pregnancy and postpartum did not increase HMO or bioactive milk proteins. No interactions or group differences in HMOs according to maternal BMI.
Larsson, et al. [19]	Maternal pre-pregnancy BMI, gestational weight gain, maternal weight at 5 and 9 months postpartum.	Prospective observational cohort study.	*n* = 60 milk samples; from 30 mother-infant dyads (13 high-weight infants and 17 normal-weight healthy Danish infants).	Maternal pre-pregnancy BMI and gestational weight gain self-reported; Maternal weight and height measured using standardized procedure at the infant’s age 5- and 9-months visits.	Well-mixed samples of right and left breasts: mothers were asked to pump the entire content of both breasts using a manual breast pump at 5 and 9 months postpartum.	HPLC after fluorescent derivatization; 19 HMOs, total HMO-bound fucose and sialic acid, total HMO. Secretor status was determined based on presence or near-absence of 2′-FL and LNFP I.	Gestational weight gain was not associated with HMO. Maternal BMI at 5 months postpartum was positively with 2′-FL, total HMO and total HMO-bound fucose; and negatively associated with 6′-SL and LSTb (all *p* ≤ 0.03). Weak associations between HMO and Maternal pre-pregnancy BMI.
McGuire et al. [20]	Maternal anthropometric indices (weight, height, BMI).	Cross-sectional, epidemiologic cohort study that involved multiple (11) international sites.	*n* = 410 milk samples; 1 from each healthy, breastfeeding woman; multisite: 40, 40, 40, 40, 40, 42, 43, 41, 24, 41, and 19 women from rural Ethiopia, urban Ethiopia, rural Gambia, urban Gambia, Ghana, Kenya, Peru, Spain, Sweden, USA Washington, and USA California.	Maternal body weight, height via questionnaire (self-reported); BMI calculated upon enrolment (2 weeks -5 months postpartum/during lactation).	1 breast only; ≤200 mL (typically 40–60 mL), manually expressed or with a breast pump; at 2 weeks–5 months postpartum.	HPLC-MS; 19 HMOs. Proportion of each HMO and the total concentration of HMOs as the sum of the annotated oligosaccharides. Secretor milk was defined as having a 2′-FL concentration that was greater than a natural, very low break in the data.	Maternal weight and BMI were positively correlated with 2′-FL (r = 0.20), FLNH (r = 0.19 and 0.15, respectively). Maternal weight was positively correlated with LNFP III (r = 0.20) and DFLNT (r = 0.14). Maternal weight and BMI were inversely correlated with LNnT and DSLNT (r = 20.16 and 20.21, respectively; and r = 20.20 and 20.24, respectively).
Moossavi et al. [29]	Maternal body composition (BMI) and fish oil supplement use during pregnancy.	Cross sectional observational study. (Representative cohort from the Longitudinal, population-based birth cohort study (CHILD)).	*n* = 393, 1 breastmilk sample from representative subset of mothers in the CHILD study (healthy term infants, Canadian multi-ethnic mothers).	BMI calculated (self-reported weight, measured height), Fish oil supplement self-reported by standardized questionnaire.	1 sample at 3–4 months postpartum (mean (SD) 17 (5) weeks postpartum), mix of foremilk and hindmilk from multiple feeds during a 24-h period; manual/hand or pump expression.	HPLC-MS; 19 HMOs, summed to estimate total HMO concentration, HMO-bound fucose (Fuc) and HMO-bound sialic acid (Sia). Maternal secretor status by the presence of 2′-FL or LNFP I.	Maternal diet and BMI are interrelated, and both can modify gut microbiota composition as well as the macro- and micro-nutrient profile and microbiota of human milk (although effect sizes were small (<2% of variation explained).
Qiao et al. [21]	Maternal dietary intake during lactation.	Cross-sectional observational study.	*n* = 90, 1 breastmilk sample per woman (healthy Chinese women with term infants (37–42 weeks)).	Validated 72-h food frequency questionnaire, weighed where possible; Chinese Dietary Reference Intakes.	1 sample taken at day 40 (±7) postpartum at the end of a breastfeed; 10 mL within 15 mins, stored at −25 °C.	HPLC-FLD; human breast milk sialic acid concentrations (free sialic acid, bound to oligosaccharides and bound to protein). No individual HMO quantification.	82.35% of the sialic acid in breastmilk was found bound to free oligosaccharides. Higher dietary intake of Vitamin A (and of milk, beef, egg, mutton, and pork) was associated with higher milk sialic acid levels (standardized coefficients = 0.713; *p* = 0.000).
Quin et al. [27]	Maternal dietary intake during lactation.	Prospective cohort clinical study.	*n* = 16 breastmilk samples, 1 per mother (healthy Euro-Canadian mothers, divided into two groups classified as milk- or almond beverage-consumers).	Self-Administered 24-h (ASA24) diet-recall survey for the 24-h period preceding milk collection.	At 5 months postpartum, manual expression of a few drops of milk (discarded) before collecting 10 mL of foremilk in the morning.	93 (median = 87) HMOs. Quantitation of total reducing sugars, Neu5Ac and Neu5Ac, neutral monosaccharide analysis (fucose, galactose), HMO profiling by CE-LIF, and targeted HMO analysis by HPLC-MS. Determination of secretor status by CE of the median levels of 2′-FL, LDFT, and LNFP I.	In Se+ samples (*n* = 12), relative levels of Fuc and Gal in HMOs were positively correlated with both the total sugars (*p* < 0.01) and total dietary fiber (*p* < 0.05) ingested within the 24-h period prior to milk collection. Several sulfonated/phosphorylated HMOs were positively correlated with breast milk monounsaturated and polyunsaturated fats, and negatively correlated with levels of saturated fats (*p* < 0.05).
Samuel et al. [8]	Maternal pre-pregnancy body composition (compared the concentrations of HMOs between overweight normal weight women at six different time points over the first four months of lactation, adjusted for milk group).	Longitudinal, observational, multicenter European study (Atlas of Human Milk Nutrients).	*n* = 1491 milk samples over 6 visits from 290 women (healthy lactating European women from France, Italy, Spain, Romania, Portugal, Sweden, and Norway).	Self-reported maternal pre-pregnancy weight and height to calculate pre-pregnancy BMI (categorized as normal weight: 18.5–24.9 kg/m2 and overweight: 25.0–29.9 kg/m2). Weight loss postpartum (kg).	Samples collected at 3 days, 14 days, 1 month, 2 months, 3 months and 4 months after delivery. Milk collected at 11h00 ± 2h00, electric breast pump, single (same) breast for the entire study (emptied in the previous feed), mixed full breast expression.	Targeted HMO analysis by HPLC-MS, and profiling of 20 HMOs by CE-LIF. Summed total amount of HMOs between milk groups, categorized mothers in one of four groups based on presence of specific α-1,2 and α-1,4- fucosylated HMOs (2′-FL, and LNFP II).	Overweight women (BMI 25.0–29.9 kg/m2) had higher concentrations of 3′SL, 6′GL (*p* < 0.05). The magnitude of the effect observed were generally low, and for 3′SL and 6′GL 22 and 29 mg/L, respectively.
Seppo et al. [22]	Maternal probiotic supplementation during pregnancy (5 × 10^9^ CFU Lactobacillus rhamnosus GG, 5 × 10^9^ CFU L. rhamnosus LC705, 2 × 10^8^ CFU Bifidobacterium breve Bb99, 2 × 10 ^9^ CFU Propionibacterium freudenreichii ssp. shermanii JS as freeze-dried capsules twice daily).	Randomized, double-blind, placebo-controlled study with 2 parallel groups (probiotic preparation or a placebo for 2 to 4 weeks before delivery, i.e., from 36 weeks’ gestation until the birth).	*n* = 81 colostrum samples (pregnant women carrying children at increased risk for allergy from the Helsinki suburban area).	Developed questionnaire to assess compliance to the supplementation.	Stored frozen colostrum samples from a previous RCT of probiotic supplementation study of 1223 pregnant women.	HPLC; 19 HMOs. Freezing does not affect HMO levels.	3FL and 3′SL significantly higher in the probiotic group (*p* = 0.008 and *p* = 0.006). Levels of DFLNH, LNnT, LNFP, 6′-SL were lower in the supplementation group (*p* = 0.005, *p* = 0.01, *p* = 0.03, *p* = 0.03). These changes are consistent with a change in select pathways in overall HMO biosynthesis.
Tonon et al. [30]	Maternal anthropometry (pre- and post-pregnancy BMI) and allergic disease status.	Cross-sectional, observational study.	*n* = 78 mature human milk samples, 1 per mother (Portuguese/Brazilian women, multi-ethnic)	Maternal pre-pregnancy BMI (self-reported weight/obtained from medical records); BMI during lactation (weight and height measured at inclusion, median: 32 days, IQR: 25–46 days postpartum); allergic disease by ISAAC questionnaire.	Manual expression of the breast opposite to the one previously emptied by the infant; 5–15 mL collected in a sterilized glass bottle in the morning (8:30–12:00 am), at 17 to 76 days postpartum (median: 32 days, IQR: 25–46 days)).	LC-MS, performed in duplicate; 16 HMOs (to represent about 90% of the total HMOs in human milk). Secretor and Lewis phenotype of the mothers based on the presence of indicative α1-2 and α1-4 fucosylated HMOs.	Maternal body composition during lactation is associated with concentrations of some HMOs in Se+ women (positive correlation between 2′-FL and maternal BMI (r = 0.30)).

FFQ, food frequency questionnaire; BMI, body mass index; HMOs, Human Milk Oligosaccharides; FUT2, Galactoside 2-alpha-L-fucosyl transferase 2; 2′-FL, 2′-Fucosyllactose; HPLC, high-performance liquid chromatography; LC-GCMS, liquid chromatography–gas chromatography–mass spectrometry; HPLC-FLD, Fluorescence detector-high performance liquid chromatography; MS, mass spectrometry; CE-LIF, Capillary electrophoresis with laser-induced fluorescence detection; Nano-HPLC-Chip/TOF MS, Nano- high-performance liquid chromatography-Chip/Time of Flight mass spectrometry; LNFP, Lacto-N-fucopentaose; LNnT, Lacto-N-Neotetraose; FM, fat mass; FFM, fat-free mass; 6′-SL, 6´Sialyllactose; LSTb, sialyl-lacto-N-tetraose.

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
