# Peer review of "Human Milk Oligosaccharide Profiles and Associations with Maternal Nutritional Factors: A Scoping Review"

_nutrients, 2021, doi:10.3390/nu13030965_

Round 1
Reviewer 1 Report
I appreciate your review for clarity and precision.
Author Response
Thank you for your review.
Revision
12 March 2021
|
Line number |
Change made |
Exact change |
|
16 |
Removed unnecessary colon |
using: changed to using |
|
19 |
Removed comma with conjunction |
criteria, changed to criteria |
|
97 |
Changed spacing after punctuation |
( maternal changed to (maternal |
|
97 |
changed punctuation placement |
OR female )AND |
|
97 |
Changed spacing after punctuation |
( diet* |
|
101 |
Changed spacing before punctuation |
OR lact* ) changed to OR lact*) |
|
102 |
Changed spacing after punctuation |
AND ( HMO changed to AND (HMO |
|
102 |
Changed spacing before punctuation |
oligosaccharide" ). Changed to oligosaccharide”). |
|
149 |
Changed spacing before punctuation |
n = 9 ), changed to (n = 9), |
|
321 |
Changed spacing before punctuation |
DFLNT ) changed to DFLNT) |
|
393 |
Removed extra spacing between words |
The majority changed to The majority |
|
450 |
Removed extra spacing between words |
known associations changed to known associations |
|
481 |
Removed extra spacing between words |
as reported changed to as reported |
|
492 |
Removed comma with conjunction |
reliable, and changed to reliable and |
|
495 |
Removed comma with conjunction |
alone, but changed to alone but |
|
495 |
Used concise language |
(in particular total changed to (total… |
|
505 |
Removed comma with conjunction |
diets, but changed to diets but |
|
519 |
Used concise language |
in particular remains changed to remains |
Reviewer 2 Report
This is a well written systematic review which contributes to the scientific field. It is a new and emerging field and this review sets out some guidance as to the further research evidence required. The abstract and introduction are informative and clearly written. The methods are detailed and clearly described. Table 1 is comprehensive and contains a lot of detail - which is good but it is difficult to understand at times due to the amount of abbreviations. I wonder if all the abbreviations and what they stand for could be listed as a footnote. Any further reduction of text in the table would also make it easier to read.
The discussion section is section is detailed but rather long. The discussion of limitations is appropriate and a balanced conclusion is provided.
Overall a high quality manuscript suitable for publication.
Author Response
Thank you for your review.
List of grammatical changes:
|
Line number |
Change made |
Exact change |
|
16 |
Removed unnecessary colon |
using: changed to using |
|
19 |
Removed comma with conjunction |
criteria, changed to criteria |
|
97 |
Changed spacing after punctuation |
( maternal changed to (maternal |
|
97 |
changed punctuation placement |
OR female )AND |
|
97 |
Changed spacing after punctuation |
( diet* |
|
101 |
Changed spacing before punctuation |
OR lact* ) changed to OR lact*) |
|
102 |
Changed spacing after punctuation |
AND ( HMO changed to AND (HMO |
|
102 |
Changed spacing before punctuation |
oligosaccharide" ). Changed to oligosaccharide”). |
|
149 |
Changed spacing before punctuation |
n = 9 ), changed to (n = 9), |
|
321 |
Changed spacing before punctuation |
DFLNT ) changed to DFLNT) |
|
393 |
Removed extra spacing between words |
The majority changed to The majority |
|
450 |
Removed extra spacing between words |
known associations changed to known associations |
|
481 |
Removed extra spacing between words |
as reported changed to as reported |
|
492 |
Removed comma with conjunction |
reliable, and changed to reliable and |
|
495 |
Removed comma with conjunction |
alone, but changed to alone but |
|
495 |
Used concise language |
(in particular total changed to (total… |
|
505 |
Removed comma with conjunction |
diets, but changed to diets but |
|
519 |
Used concise language |
in particular remains changed to remains |
Regarding Table 1: abbreviations consistent throughout and where possible, duplication of names and their abbreviations were removed:
For example,
Levels of difucosyllacto-N-hexaose, lacto-N-tetraose, lacto-N-fucopentaose and 6′-sialyllactose 6’ changed to DFLNH, LNnT, LNFP, 6’-SL (Seppo et al).
We have also included the footnote:
Reviewer 3 Report
This is a large work of extensive reviews on human milk oligossacharides published between 2013 and March 2020.
14 publications were included in the scope review, with data from 1,388 lactating women and 4,011 milk samples . Reported articles on maternal dietary intake (n = 3), dietary supplementation interventions (n = 2) and maternal body composition (n = 9), HMO relation composition of milk samples. The articles reflect a fairly recent interest in the relationship between maternal factors and HMO composition.
Author Response
Thank you for your review.
Revision
12 March 2021
|
Line number |
Change made |
Exact change |
|
16 |
Removed unnecessary colon |
using: changed to using |
|
19 |
Removed comma with conjunction |
criteria, changed to criteria |
|
97 |
Changed spacing after punctuation |
( maternal changed to (maternal |
|
97 |
changed punctuation placement |
OR female )AND changed to OR female) |
|
97 |
Changed spacing after punctuation |
( diet* changed to (diet* |
|
101 |
Changed spacing before punctuation |
OR lact* ) changed to OR lact*) |
|
102 |
Changed spacing after punctuation |
AND ( HMO changed to AND (HMO |
|
102 |
Changed spacing before punctuation |
oligosaccharide" ). Changed to oligosaccharide”). |
|
149 |
Changed spacing before punctuation |
n = 9 ), changed to (n = 9), |
|
321 |
Changed spacing before punctuation |
DFLNT ) changed to DFLNT) |
|
393 |
Removed extra spacing between words |
The majority changed to The majority |
|
450 |
Removed extra spacing between words |
known associations changed to known associations |
|
481 |
Removed extra spacing between words |
as reported changed to as reported |
|
492 |
Removed comma with conjunction |
reliable, and changed to reliable and |
|
495 |
Removed comma with conjunction |
alone, but changed to alone but |
|
495 |
Used concise language |
(in particular total changed to (total… |
|
505 |
Removed comma with conjunction |
diets, but changed to diets but |
|
519 |
Used concise language |
in particular remains changed to remains |